# D-2-Hydroxyglutarate Attenuates Sinonasal Inflammation in Murine Allergic Rhinitis

**Anuj Tharakan, Ankit Kumar, Carmen Camarena** (ID)**, Daniel H. Conrad and Rebecca K. Martin** *(ID)

Department of Microbiology and Immunology, School of Medicine, Virginia Commonwealth University, Richmond, VA 23298, USA; tharakana@vcu.edu (A.T.); akumar6@vcu.edu (A.K.); camarenam@vcu.edu (C.C.); daniel.conrad@vcuhealth.org (D.H.C.)
* Correspondence: rebecca.martin@vcuhealth.org

**Abstract:** Introduction: Allergic rhinitis (AR) is largely driven by IgE-induced immune cell activation, which promotes allergen-induced upper airway inflammation. The regulatory mechanisms of IgE synthesis in AR are poorly understood. Several analyses associate single nucleotide polymorphisms (SNPs) which reduce the expression of the *D2HGDH* gene with AR. *D2HGDH* encodes an enzyme that converts D-2-hydroxyglutarate (D2HG) to α-ketoglutarate (α-KG). This study aims to clarify the relationship between AR and SNPs in *D2HGDH*. Methods: Mice were treated with vehicle control or octyl-D2HG prior to intranasal exposure to *Alternaria alternata*. Draining lymph nodes (dLNs) were then evaluated for IgE-producing cells and T-cell polarization. Next, mice were exposed to intranasal Alternaria on days 0, 10, 20, and 27–30 and were treated intranasally with octyl-D2HG or vehicle control on days 20 and 27. Nasal inflammation was analyzed in nasal lavage fluid (NLF) cellularity and antigen-specific IgE production. Results: The administration of D2HG prior to Alternaria exposure suppressed IgE synthesis ($p < 0.01$) and Th2 cell polarization ($p < 0.01$) in dLNs. In a murine model of AR, D2HG administration reduced overall cellular infiltrates and eosinophils in NLF. Further, antigen-specific IgE in NLF was significantly reduced in mice treated with D2HG ($p < 0.05$). Conclusions: An analysis of the regulatory landscape surrounding the rs34290285 SNP demonstrates that the downregulation of *D2HGDH* expression reduces the risk of AR. Downregulation of *D2HGDH* likely results in accumulation of D2HG intracellularly, suggesting that D2HG is protective against allergic rhinitis. We show that the administration of D2HG impairs IgE production, leading to the amelioration of allergic sinonasal inflammation in a murine model of AR. These findings suggest a causal relationship between *D2HGDH* expression, D2HG levels, and allergic rhinitis risk.

**Keywords:** D-2-hydroxyglutarate; allergy; IgE; rhinitis; immunometabolism

## 1. Introduction

Allergic rhinitis (AR) is a highly prevalent global health problem affecting 20–30% of adults worldwide [1]. AR is a clinical syndrome characterized by chronic allergic inflammation of the nasal mucosa, which results in symptoms of congestion, sneezing, itching, and rhinorrhea [2]. AR is caused by a combination of T helper 2 cytokine production, such as IL-4, IL-5, and IL-13, as well as IgE-mediated inflammatory mechanisms [3]. Though these processes that propagate allergic inflammation have been well studied, the mechanisms underlying the initiation of these inflammatory mechanisms remain poorly understood.

Recent studies have highlighted the role of cellular metabolism as a critical regulator of immune function [4,5]. These studies have demonstrated that alterations in cellular

metabolism by antigen-presenting cells and dendritic cells can result in profound changes in adaptive immune responses in vivo [6,7]. Our group and others have previously identified that single nucleotide polymorphisms (SNPs) in the *D2HGDH* gene have been associated with multiple allergic disorders, including AR [8–10]. The SNP rs34290285 demonstrates that the downregulation of *D2HGDH* expression reduces the risk of AR [8–10]. *D2HGDH* encodes an enzyme that is responsible for the conversion of the metabolite D-2-hydroxyglutarate (D2HG) to $\alpha$-ketoglutarate, and the downregulation of *D2HGDH* results in the accumulation of D2HG [11,12]. These metabolites are critical endogenous regulators of the 2-oxoglutarate-dependent family enzymes, with $\alpha$-ketoglutarate acting as a substrate and D2HG acting as a competitive inhibitor for these enzymes [11]. Our group has previously shown that administration of D2HG in murine models of allergic asthma suppresses type 2 inflammation and local IgE synthesis [8].

In this study, we sought to assess the effects of D2HG on sinonasal allergic inflammation in a murine model of *Alternaria alternata*-induced allergic rhinitis. We demonstrate that the administration of a cell-permeable form of D2HG prior to intranasal exposure to *Alternaria* results in reduced IgE responses in the submandibular lymph node. Additionally, D2HG administration to mice with established allergic rhinitis resulted in significant suppression of antigen-specific IgE levels and amelioration of sinonasal allergic inflammation.

## 2. Methods

### 2.1. Mice

C57Bl/6J mice were purchased from the Jackson Laboratory. Mice were housed under specific-pathogen-free conditions in the Virginia Commonwealth University (VCU) barrier vivarium facility in accordance with the humane treatment of laboratory animals set forth by the National Institutes of Health. Animal protocols were conducted under the permission and with the oversight of the VCU Institutional Animal Care and Use Committee. Mice were randomized for experiments with equal numbers of male and female mice used unless otherwise specified. Experimental mouse ages were between 6 and 12 weeks old. Mice were anesthetized with isoflurane and intranasally instilled with the indicated stimuli in a 20 µL volume, where indicated. Murine allergic rhinitis models were induced via intranasal administration of 10 µg Alternaria (Greer Laboratories, Lenoir, NC, USA) + 100 µg OVA (Sigma-Aldrich, Burlington, MA, USA) on days 0 and 10. Mice were then treated intranasally with 5 µg *Alternaria* + 50 µg OVA on days 20 and 27–30 and analyzed on day 31. Where indicated, mice were treated with 100 µg of octyl-D2HG (Cayman Chemical, Ann Arbor, MI, USA) one hour prior to *Alternaria* + OVA exposure.

### 2.2. Flow Cytometry

Antibodies used for flow cytometric analysis were TCRβ (clone: H57-597), CD4 (clone: GK1.5), CD44 (clone: IM7), PD-1 (clone: 29F.1A12), CXCR5 (clone: L138D7), IL-4 (clone: 11B11), IL-5 (clone: TRFK5), FoxP3 (clone: MF-14), B220 (clone: RA3-6B2), CD138 (clone: 281-2), CD95 (clone: Jo2), GL7 (clone: GL7), IgG1 (clone: A85-1), IgE (clone: RME-1), Gr-1 (clone: RB6-8C5), Siglec-F (clone: E50-2440). Viability staining was performed with Zombie Aqua Fixable Viability Dye (Biolegend). Cells were stained with viability dye according to the manufacturer's instructions and blocked with anti-CD16/32 (clone 2.4g2) prior to staining on ice with antibodies for surface molecules. Cells were then analyzed on a LSRFortessa (BD Biosciences, San Jose, CA, USA). Intracellular cytokine staining was performed after incubation for 4h with 1 µM ionomycin (Sigma-Aldrich), 1 µM 1 phorbol 12-myristate 13-acetate (Sigma-Aldrich), and Brefeldin A (Biolegend, San Diego, CA, USA). Cells were then fixed and stained using Fixation buffer (Biolegend) and Intracellular Staining Permeabilization Wash Buffer (Biolegend) per the manufacturer's

instructions. Transcription factor staining was performed by fixation and permeabilization using the True-Nuclear Transcription Factor Buffer Set (Biolegend). Different cell types were identified by the following gating strategies: T follicular helper type 2 (Tfh2) cells (Live, $TCR\beta^+$, $CD4^+$, $B220^-$, $CD44^+$, $PD-1^+$, $CXCR5^+$, $IL-4^+$), Germinal center B cells (GCBC) (Live, $B220^+$, $TCR\beta^+$, $CD138^-$, $CD95^+$, $GL7^+$), Tfr cells (Live, $TCR\beta^+$, $CD4^+$, $B220^-$, $CD44^+$, $PD-1^+$, $CXCR5^+$, $FoxP3^+$), Plasma cells (Live, $B220^+$, $TCR\beta^-$, $CD138^+$), eosinophils (Live, $CD45^+$, $CD11b^+$, $CD11c^{lo}$, $Siglec-F^+$, $SSC^{hi}$), neutrophils (Live, $CD45^+$, $Siglec-F^-$, $CD11c^{lo}$, $CD11b^+$, $Ly6G^+$), basophils (Live, $CD45^+$, $CD11c^{lo}$, $Siglec-F^-$, $Fc\varepsilon RI\alpha^+$, $c-Kit^-$), mast cells (Live, $CD45^+$, $CD11c^{lo}$, $Siglec-F^-$, $Fc\varepsilon RI\alpha^+$, $c-Kit^+$).

### 2.3. Staining for IgE B Cells

$IgE^+$ B cell staining was performed as previously described [13]. Lymph node cell suspensions were incubated on ice for 10 min with viability dye and 2.4g2 Fc Block. Surface staining was performed with the addition of unlabeled anti-IgE (clone RME-1) to saturate Fc receptor-bound IgE. Cells were then fixed with and permeabilized with a BD intracellular staining kit and stained with fluorochrome-conjugated anti-IgE (clone RME-1) and anti-IgG1 overnight at 4 °C.

### 2.4. Nasal Lavage and Cytology

Transpharyngeal nasal lavage fluid (NLF) was collected as previously described [14]. Nasal lavage fluid was centrifuged at $500\times g$ for 5 min. Cell pellets from lavage samples were resuspended in 100 μL PBS with 2 mM EDTA, and total cells were prepared for flow cytometric analysis as described above.

### 2.5. ELISA

OVA-specific IgE ELISA was performed as previously described [15]. Briefly, plates were coated with 5 μg/mL anti-(R1E4), blocked, and incubated with diluted serum samples. Detection was performed with biotinylated OVA followed by streptavidin-HRP (ThermoFisher, Waltham, MA, USA). Plates were developed with tetramethylbenzidine (BD Biosciences) and stopped with sulfuric acid before reading at 450 nm on a microplate reader (Molecular Devices, San Jose, CA, USA). A sandwich ELISA for total IgE at a concentration of 10 ng/mL was used as a reference standard to calculate arbitrary units.

### 2.6. Statistical Analysis

The details of the replicates for each experiment are listed in the figure legends. In all experiments, data are shown as mean $\pm$ SEM, and statistical analyses were performed using GraphPad Prism software (version 8.2.1). An unpaired Student's *t* test, one-way ANOVA with Tukey's multiple comparisons post-test, or the Mann–Whitney test was applied for significance testing as indicated. Statistical significance is defined as * $p < 0.05$, ** $p < 0.01$, *** $p < 0.001$, **** $p < 0.0001$, or NS (not significant).

## 3. Results

### 3.1. D2HG Suppresses Allergic Sensitization In Vivo

To evaluate the effect of D2HG on allergic sensitization in AR, mice were treated intranasally with 100 μg of octyl-D2HG, a cell-permeable precursor of D2HG [16], one hour prior to intranasal exposure to *Alternaria* + OVA. An analysis of T-cell responses in the submandibular lymph node 8 days after exposure revealed a significant reduction in Th2 cell and Tfh2 cell polarization in mice treated with octyl-D2HG compared to vehicle controls (Figure 1A–C).

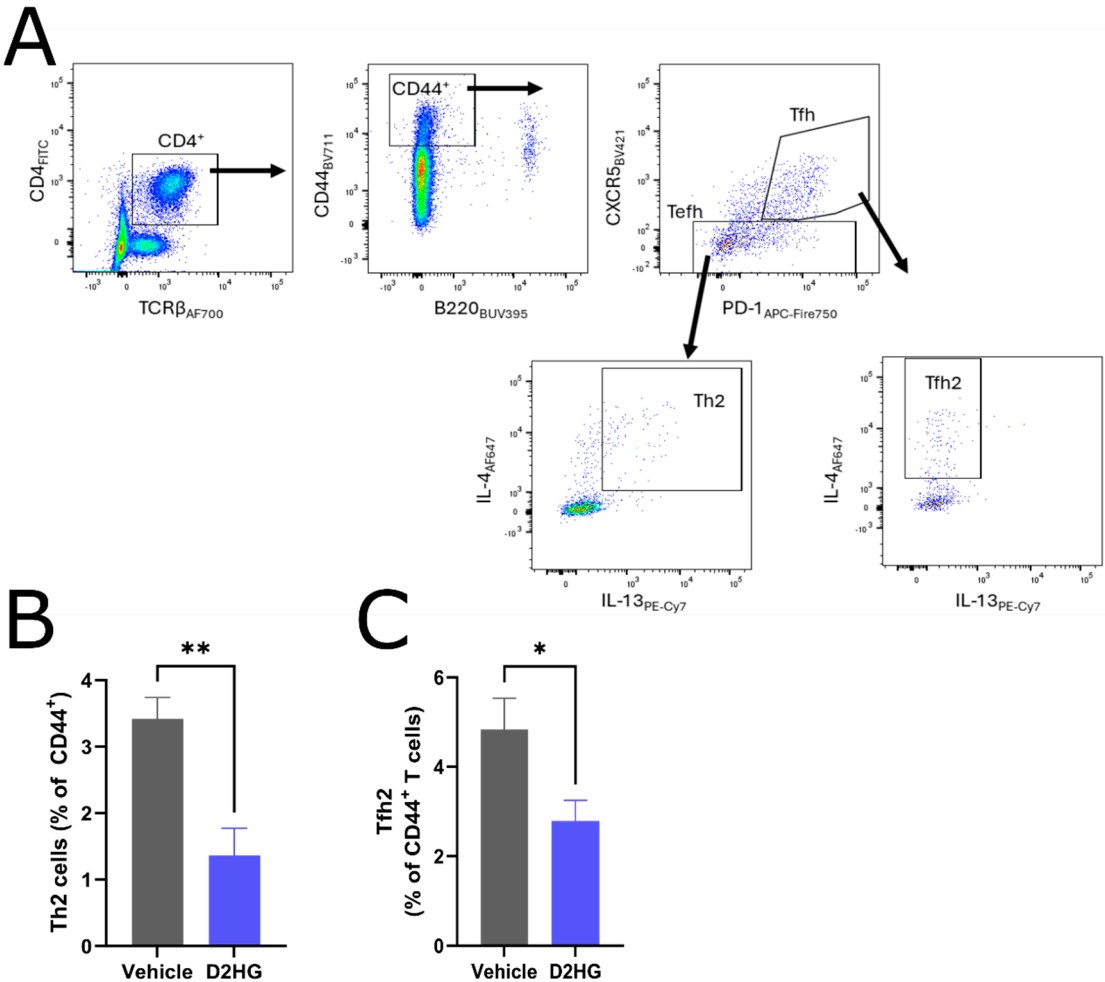

**Figure 1.** D2HG inhibits Th2 and Tfh2 cell polarization in vivo. Mice were treated intranasally with octyl-D2HG one hour prior to *Alternaria* + OVA exposure, and submandibular lymph nodes were analyzed for T-cell polarization by flow cytometry. (**A**) Representative gating strategy from a control sample for populations graphed in (**B**,**C**). (**B**) Th2 cells and (**C**) Tfh2 cells were quantified. Data are presented as mean $\pm$ SEM. N = 8–12 per group. * $p < 0.05$, ** $p < 0.01$.

### 3.2. IgE Synthesis in Draining Lymph Nodes Is Impaired by D2HG Administration

As Tfh2 cells are critical for IgE responses, the effect of D2HG on IgE synthesis was evaluated by analyzing B cells in the submandibular lymph node following *Alternaria* + OVA exposure (Figure 2A). Mice treated with octyl-D2HG prior to *Alternaria* + OVA exposure exhibited a significant reduction in IgE$^+$ B cells compared to control mice (Figure 2B). These mice also demonstrate a substantial reduction in IgG1$^+$ B cells in the submandibular lymph node (Figure 2C).

### 3.3. D2HG Ameliorates Sinonasal Allergic Inflammation in Murine AR

To assess the efficacy of D2HG administration as a potential AR therapy, we tested the effect of D2HG treatment in a murine model of AR. Mice were sensitized intranasally with Alternaria + OVA on day 0, boosted on day 10, and challenged on days 17–21. Mice were randomized to receive treatment with intranasal octyl-D2HG or vehicle control one hour prior to Alternaria + OVA exposures on days 10 and 17 (Figure 3A). On day 22, mice were evaluated for sinonasal inflammation by assessing NLF cellularity. Mice treated with octyl-D2HG exhibited a significant reduction in total NLF cells and a modest reduction in NLF eosinophils (Figure 3B,C). Further, measurement of OVA-specific IgE levels revealed that octyl-D2HG-treated mice had significant reductions in antigen-specific IgE production (Figure 4).

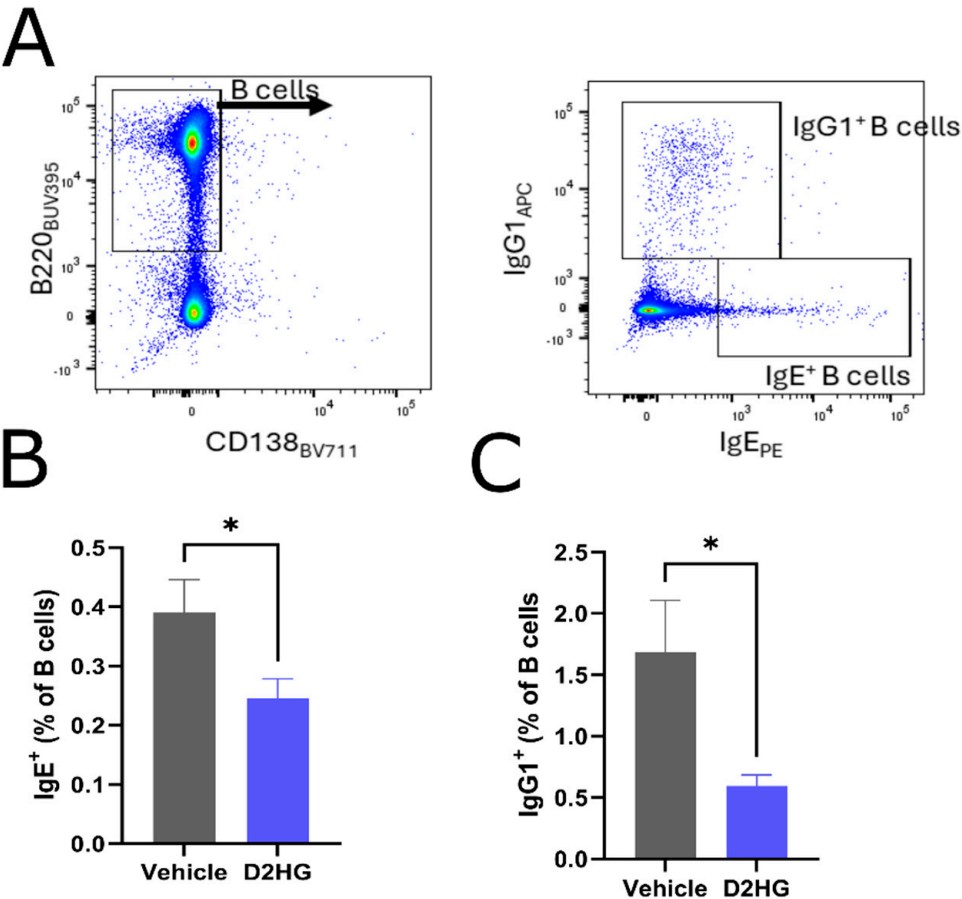

**Figure 2.** D2HG treatment reduces IgE synthesis in draining lymph nodes. Mice were treated intranasally with octyl-D2HG one hour prior to Alternaria + OVA exposure, and submandibular lymph nodes were analyzed for IgE+ and IgG1+ B cells by flow cytometry. (**A**) Representative gating strategy for populations graphed in (**B**,**C**). (**B**) IgE+ B cell data are presented as mean ± SEM. N = 8–12 per group. * $p < 0.05$.

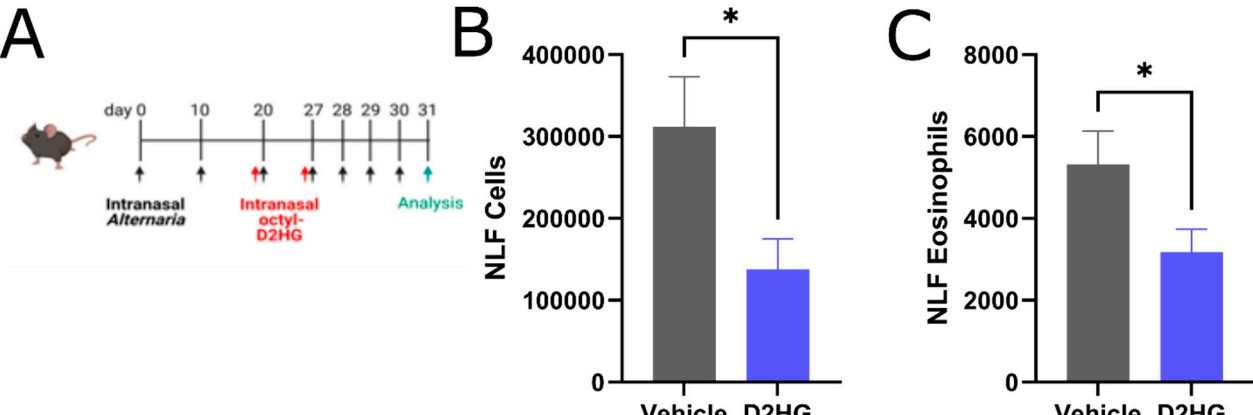

**Figure 3.** Sinonasal inflammatory infiltrates are reduced in mice treated with D2HG. (**A**) Mice were sensitized with Alternaria + OVA on days 0, 10, and 20, and challenged on days 27–30. Mice were treated with intranasal D2HG on days 20 and 27. (**B**) Mice were analyzed on day 31 for total cells and (**C**) eosinophils (Live, Siglec-F+, CD11c-) in NLF by flow cytometry. Data are presented as mean ± SEM. N = 8–12 per group. * $p < 0.05$.

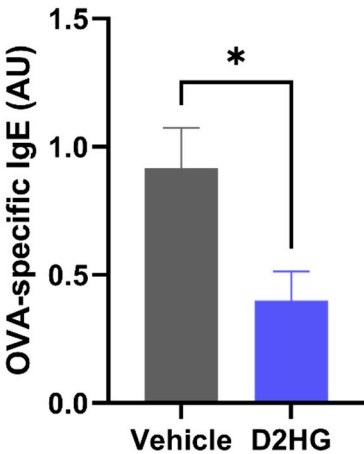

**Figure 4.** Mice treated with D2HG exhibit decreased production of antigen-specific IgE. AR model and D2HG treatments were performed as in Figure 3. Serum was collected on day 31 and analyzed for OVA-specific IgE levels by ELISA. Data are presented as mean $\pm$ SEM. N = 8–12 per group. * $p < 0.05$.

## 4. Discussion

Sinonasal inflammatory disorders such as CRSwNP and AR affect up to 30% of the global population [1]. These disorders are typically characterized by type 2 inflammatory infiltration in the nasal and paranasal sinus mucosa [3]. Type 2 inflammation, typified by eosinophilic cellular infiltrates, is mediated by a combination of the Th2 cytokines IL-4, IL-5, and IL-13 and antigen-specific IgE production [3]. Previous studies have demonstrated that the underlying etiology of these type 2 inflammatory disorders is likely a combination of underlying genetic predisposition and environmental exposures [17,18].

Here, we explore the potential consequences of an SNP in the *D2HGDH* gene, which has been previously implicated in allergic disorders such as allergic asthma, food allergy, atopic dermatitis, and allergic rhinitis [8–10]. Previous studies by our group and others have indicated that the polymorphisms in this gene that reduce the risk of allergic disease result in reduced *D2HGH* expression [9,10]. *D2HGDH* encodes for an enzyme that converts D2HG to $\alpha$-ketoglutarate [11]. This suggests that SNPs that reduce *D2HGH* expression, and therefore lead to cellular accumulation of D2HG, reduce the risk of allergic disease.

Here, we demonstrate the local administration of D2HG prior to initial nasal allergen exposure impairs Th2 and Tfh2 polarization and suppresses IgE synthesis in vivo. Further, we show that D2HG treatment reduces sinonasal type 2 inflammation and antigen-specific IgE production in a murine model of AR. These findings further support a causal role for D2HG, and SNPs in the *D2HGDH* gene, in the regulation of allergic sensitization in type 2 sinonasal inflammatory disorders like AR and CRSwNP. Our study demonstrates that D2HG reduces IgE levels in serum; however, local IgE synthesis and release in the nasal mucosa have also been implicated in the pathogenesis of allergic rhinitis. An analysis of local class switch recombination and IgE release in the nasal tissues will be an important avenue for future research. Additionally, we demonstrate the efficacy of D2HG in a therapeutic modality, suggesting that D2HG or its downstream targets may have promise as novel therapeutics in these disorders.

D2HG and $\alpha$-ketoglutarate reciprocally regulate a family of enzymes known as 2-oxoglutarate-dependent enzymes (2-OGDDs) [19]. The enzymes in this family utilize $\alpha$-ketoglutarate as a substrate and are competitively inhibited by D2HG [20]. These enzymes include epigenetic modulators such as the ten-eleven translocation enzymes, which are DNA demethylases, the Jumonji-domain containing enzymes, which are histone demethylases, and prolyl hydroxylase enzymes which regulate HIF-1$\alpha$ levels and collagen synthesis [21]. Thus, D2HG may influence allergic sensitization in type 2 inflammatory

disorders through any combination of these downstream mechanisms. Further work is needed to better clarify the epigenetic and transcriptional effects by which D2HG affects the initiation of type 2 immune responses.

## 5. Conclusions

D2HG treatment prior to initial intranasal exposure to an allergen significantly abrogated allergic sensitization as measured by Th2 and Tfh2 cell polarization and primary IgE synthesis. Further, treatment with D2HG therapeutically in established murine AR promoted significant reductions in sinonasal allergic inflammation and antigen-specific IgE production, indicating that D2HG or its downstream targets may represent a viable novel therapeutic approach for type 2 inflammatory disorders of the upper airway.

**Author Contributions:** Conceptualization, A.T. and R.K.M.; methodology, A.T.; software, formal analysis, A.T.; data curation, A.T., A.K. and C.C.; writing—review and editing, A.T., D.H.C. and R.K.M.; supervision, R.K.M.; project administration, R.K.M.; funding acquisition, R.K.M. All authors have read and agreed to the published version of the manuscript.

**Funding:** This work was supported by funding from VCU's CTSA (UL1TR002649 from the National Center for Advancing Translational Sciences) to R.K.M. and the CCTR Endowment Fund of VCU as well as funding from NHLBI-NIH (R01HL162991) to RKM. Services and products in support of the research project were generated by the Virginia Commonwealth University (VCU) Massey Comprehensive Cancer Center Flow Cytometry Shared Resource, supported, in part, by funding from NIH-NCI Cancer Center Support Grant P30 CA016059.

**Data Availability Statement:** Raw data is available upon request.

**Conflicts of Interest:** A.T. and R.K.M. hold stock in Pleros Therapeutics which is currently of no value.

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
