# Peer review of "D-2-Hydroxyglutarate Attenuates Sinonasal Inflammation in Murine Allergic Rhinitis"

_allergies, doi:10.3390/allergies5020013_

Round 1

Reviewer 1 Report

Comments and Suggestions for Authors

This study demonstrates that D2HG suppresses Th2 and Tfh2 cell differentiation and IgE production in allergic rhinitis, thereby alleviating allergic inflammation and suggesting the potential of D2HG as a novel therapeutic approach. In particular, the study highlights the immunomodulatory effects of D2HG, suggesting its value as a new drug candidate for allergic diseases. While this is an intriguing study, I have the following concerns:

1.         Regarding Figures 1A and 2A, the flow cytometry results of D2HG-treated mice lack clarification as to whether the data presented were selected as representative examples, and there is insufficient explanation for their interpretation. Presentation of typical flow cytometry dot plot data for both the control (untreated) and D2HG-treated groups in the figures would make the differences between the two groups more visually apparent.

2.         The text mentions that 100 micrograms of D2HG was administered intranasally, but there is no adequate explanation of the rationale for this dosage or the estimated local concentration achieved. What is the basis for selecting this dosage, and what is the estimated local concentration achieved in the nasal cavity with a 100-microgram administration?

3.         It is hypothesized that octyl-D2HG is degraded intracellularly to D2HG, which produces the observed effects, but no specific evidence or experimental data are provided to support this claim. Without clarification, it remains unclear whether the observed effects are due to D2HG itself or whether octyl-D2HG directly exerts the effects. Experimental data or references supporting the degradation of octyl-D2HG to D2HG should be provided.

4.         There appears to be a lack of critical control groups necessary to determine whether D2HG acts specifically on allergic rhinitis. In particular, a group receiving D2HG without exposure to Alternaria has not been included, making it unclear whether the effects of D2HG are dependent on allergen induction or reflect a broader anti-inflammatory effect. For example, it would be necessary to include a group treated with D2HG alone and assess eosinophil count, T cell differentiation, and IgE production. In addition, evaluation of its effects in non-allergic inflammation models (e.g., LPS-induced models) and comparisons with other allergens could help clarify whether the effect of D2HG is specific to allergic inflammation.

Author Response

Comments from Reviewer 1:

This study demonstrates that D2HG suppresses Th2 and Tfh2 cell differentiation and IgE production in allergic rhinitis, thereby alleviating allergic inflammation and suggesting the potential of D2HG as a novel therapeutic approach. In particular, the study highlights the immunomodulatory effects of D2HG, suggesting its value as a new drug candidate for allergic diseases. While this is an intriguing study, I have the following concerns:

  1. Regarding Figures 1A and 2A, the flow cytometry results of D2HG-treated mice lack clarification as to whether the data presented were selected as representative examples, and there is insufficient explanation for their interpretation. Presentation of typical flow cytometry dot plot data for both the control (untreated) and D2HG-treated groups in the figures would make the differences between the two groups more visually apparent.

-We have clarified this in the figure legend that this is a representative gating strategy from a control sample.

  1. The text mentions that 100 micrograms of D2HG was administered intranasally, but there is no adequate explanation of the rationale for this dosage or the estimated local concentration achieved. What is the basis for selecting this dosage, and what is the estimated local concentration achieved in the nasal cavity with a 100-microgram administration?

-While we do not have the local concentration achieved in the nasal cavity and it would be beyond the scope of this work to determine tissue drug concentrations, we carefully decided this dose using an intranasal dose response curve that is published in our previous work (PMID: 37408337). In this manuscript we utilized 50 and 250ug. As 50ug was not significantly different from control in the lung, but 250ug was in the asthma model, we decided to utilize 100ug in these studies examining allergic rhinitis.

  1. It is hypothesized that octyl-D2HG is degraded intracellularly to D2HG, which produces the observed effects, but no specific evidence or experimental data are provided to support this claim. Without clarification, it remains unclear whether the observed effects are due to D2HG itself or whether octyl-D2HG directly exerts the effects. Experimental data or references supporting the degradation of octyl-D2HG to D2HG should be provided.

-We have now added another reference that illustrates that Octyl-D2HG is the cell permeable precursor to D2HG. (PMID: 31151327).

  1. There appears to be a lack of critical control groups necessary to determine whether D2HG acts specifically on allergic rhinitis. In particular, a group receiving D2HG without exposure to Alternaria has not been included, making it unclear whether the effects of D2HG are dependent on allergen induction or reflect a broader anti-inflammatory effect. For example, it would be necessary to include a group treated with D2HG alone and assess eosinophil count, T cell differentiation, and IgE production. In addition, evaluation of its effects in non-allergic inflammation models (e.g., LPS-induced models) and comparisons with other allergens could help clarify whether the effect of D2HG is specific to allergic inflammation.

-We have utilized D2HG in allergic asthma models and seen similar results. The turnaround time for this review dose not allow for the repetition of all of our experiments. We have included a vehicle control which is customary in similar publications. Adding other models to this manuscript with LPS, is outside the scope of this manuscript.

Reviewer 2 Report

Comments and Suggestions for Authors

D-2-hydroxyglutarate attenuates sinonasal inflammation in mu-rine

It is a nice and original study conducted in mice sensitized to allergic rhinitis towards alternaria

The methodology seems correct and the authors tend to demonstrate the efficacy of that molecule to prevent allergic rhinitis and to decrease the symptoms presumaly but objectively it seems to decrease toe level of Ig E and nasal inflammation

The molecule they use seem to decreases the inflammtion whne it is administered through the nose

The authors say that this prevents the onset of an allergic rhinitis

Question is this molecule currently availbal on the market?

Are the results trasposable in a clinical study with patients;

is the molecule specifically indicated to decrease an Ig E mediator reaction?

Is it ok for asthma? You have already published a study about asthama

What is new with this study on allergic rhinitis

When we know that the pathgenesis is not completely understood what would be the palce of your molecule compared to the biologicals?anti IgE, anti IL5, anti IL4 and 13?

The methodology of the study seems ok

The applicability of this study in humen is questionable

Good originality but poor impact for the clinic

Author Response

Comments from Reviewer 2:

D-2-hydroxyglutarate attenuates sinonasal inflammation in murine.It is a nice and original study conducted in mice sensitized to allergic rhinitis towards alternaria. The methodology seems correct and the authors tend to demonstrate the efficacy of that molecule to prevent allergic rhinitis and to decrease the symptoms presumably but objectively it seems to decrease the level of IgE and nasal inflammation. The molecule they use seem to decreases the inflammation when it is administered through the nose. The authors say that this prevents the onset of an allergic rhinitis.

Question is this molecule currently available on the market?

-This molecule is not currently in clinical trials, but is available for purchase for research purposes.

Are the results transposable in a clinical study with patients;

-It has not been utilized in patients, but there are indications that link it to human disease and SNPs in the enzyme D2HGDH. See our previous reference (PMID: 37408337).

is the molecule specifically indicated to decrease an Ig E mediator reaction?

-As we have found that this molecule reduces IgE, we assume that downstream IgE-mediated Mast cell degranulation is reduced. We see evidence of this in our asthma model paper (PMID: 37408337).

Is it ok for asthma? You have already published a study about asthma

-Yes, we have found it works effectively to reduce asthma in murine models and immune responses in the lung.

What is new with this study on allergic rhinitis

-This study looks at local allergic rhinitis and local immune infiltration and IgE responses in the NSF.

When we know that the pathogenesis is not completely understood what would be the place of your molecule compared to the biologicals?anti IgE, anti IL5, anti IL4 and 13?

-We have not benchmarked it to other drugs used in human allergic diseases. From the work in this manuscript, D2HG is working upstream of IgE production (as it shows an effect on T cell polarization).

The methodology of the study seems ok

The applicability of this study in human is questionable

-This is a great comment and human work needs to be done. This is beyond the scope of this paper.

Good originality but poor impact for the clinic

Reviewer 3 Report

Comments and Suggestions for Authors

The study by Tharakan et al. is properly designed and the articel correctly structured and written. The introduction gives a short yet complete account of hitherto performer studies and justifies tackling on the assessment of D-2-hydroxyglutarate infilence on nasal allergic inflammation in the murine model.

Results are addressed in the discussion and are commented on in the context of possible mechanisms leading to presented findings.

I have some minor suggestions with regard ti the manuscript:

1.        In the methods section, please clarify that specific IgE’s were measured in serum. This is not clearly stated and some may get confused whether IgE was assessed in serum or in the nasal lavage fluid, which is often the case in human rhinitis studies.

2.        Could you comment/speculate if Alternaria-specific IgE assessment in that model would be feasible?

3.        Is the assessment of IgE specific for OVA and/or Alternaria in the NLF and not only in serum justified or planned in further studies? Some comments regarding this would enrich the discussion.

4.        References should be formatted to conform with the format recommended by the publishing house.

Author Response

Comments from Reviewer 3:

The study by Tharakan et al. is properly designed and the article correctly structured and written. The introduction gives a short yet complete account of hitherto performer studies and justifies tackling on the assessment of D-2-hydroxyglutarate influence on nasal allergic inflammation in the murine model.

Results are addressed in the discussion and are commented on in the context of possible mechanisms leading to presented findings.

I have some minor suggestions with regard to the manuscript:

  1. In the methods section, please clarify that specific IgE’s were measured in serum. This is not clearly stated and some may get confused whether IgE was assessed in serum or in the nasal lavage fluid, which is often the case in human rhinitis studies.

- This has now been corrected in the manuscript.

  1. Could you comment/speculate if Alternaria-specific IgE assessment in that model would be feasible?

-Alternaria-specific IgE can be measured, but is much more difficult as Alternaria is a mixture of antigens. Using the bystander protein as a molecular tool is a common technique in allergy research to measure antigen-specific Ig.

  1. Is the assessment of IgE specific for OVA and/or Alternaria in the NLF and not only in serum justified or planned in further studies? Some comments regarding this would enrich the discussion.

-A comment was added to the discussion.

  1. References should be formatted to conform with the format recommended by the publishing house.

-We have updated these

Round 2

Reviewer 1 Report

Comments and Suggestions for Authors

I am in agreement with the acceptance of this paper as the authors have adequately addressed the concerns of my peer review.
